# Stochastic Optimization with Variance Reduction for Infinite Datasets with Finite Sum Structure

**Alberto Bietti**
Inria[*]
alberto.bietti@inria.fr

**Julien Mairal**
Inria[*]
julien.mairal@inria.fr

## Abstract

Stochastic optimization algorithms with variance reduction have proven successful for minimizing large finite sums of functions. Unfortunately, these techniques are unable to deal with stochastic perturbations of input data, induced for example by data augmentation. In such cases, the objective is no longer a finite sum, and the main candidate for optimization is the stochastic gradient descent method (SGD). In this paper, we introduce a variance reduction approach for these settings when the objective is composite and strongly convex. The convergence rate outperforms SGD with a typically much smaller constant factor, which depends on the variance of gradient estimates only due to perturbations on a *single* example.

## 1 Introduction

Many supervised machine learning problems can be cast as the minimization of an expected loss over a data distribution with respect to a vector $x$ in $\mathbb{R}^p$ of model parameters. When an infinite amount of data is available, stochastic optimization methods such as SGD or stochastic mirror descent algorithms, or their variants, are typically used (see [5, 11, 24, 34]). Nevertheless, when the dataset is finite, incremental methods based on variance reduction techniques (*e.g.*, [2, 8, 15, 17, 18, 27, 29]) have proven to be significantly faster at solving the finite-sum problem

$$\min_{x \in \mathbb{R}^p} \left\{ F(x) := f(x) + h(x) = \frac{1}{n} \sum_{i=1}^{n} f_i(x) + h(x) \right\}, \tag{1}$$

where the functions $f_i$ are smooth and convex, and $h$ is a simple convex penalty that need not be differentiable such as the $\ell_1$ norm. A classical setting is $f_i(x) = \ell(y_i, x^\top \xi_i) + (\mu/2)\|x\|^2$, where $(\xi_i, y_i)$ is an example-label pair, $\ell$ is a convex loss function, and $\mu$ is a regularization parameter.

In this paper, we are interested in a variant of (1) where random perturbations of data are introduced, which is a common scenario in machine learning. Then, the functions $f_i$ involve an expectation over a random perturbation $\rho$, leading to the problem

$$\min_{x \in \mathbb{R}^p} \left\{ F(x) := \frac{1}{n} \sum_{i=1}^{n} f_i(x) + h(x) \right\}. \quad \text{with} \quad f_i(x) = \mathbb{E}_\rho[\tilde{f}_i(x, \rho)]. \tag{2}$$

Unfortunately, variance reduction methods are not compatible with the setting (2), since evaluating a single gradient $\nabla f_i(x)$ requires computing a full expectation. Yet, dealing with random perturbations is of utmost interest; for instance, this is a key to achieve stable feature selection [23], improving the generalization error both in theory [33] and in practice [19, 32], obtaining stable and robust predictors [36], or using complex a priori knowledge about data to generate virtually

---

[*]Univ. Grenoble Alpes, Inria, CNRS, Grenoble INP, LJK, 38000 Grenoble, France

Table 1: Iteration complexity of different methods for solving the objective (2) in terms of number of iterations required to find $x$ such that $\mathbb{E}[f(x) - f(x^*)] \leq \epsilon$. The complexity of N-SAGA [14] matches the first term of S-MISO but is asymptotically biased. Note that we always have the perturbation noise variance $\sigma_p^2$ smaller than the total variance $\sigma_{\text{tot}}^2$ and thus S-MISO improves on SGD both in the first term (linear convergence to a smaller $\bar{\epsilon}$) and in the second (smaller constant in the asymptotic rate). In many application cases, we also have $\sigma_p^2 \ll \sigma_{\text{tot}}^2$ (see main text and Table 2).

| Method | Asymptotic error | Iteration complexity |
|--------|------------------|----------------------|
| SGD | 0 | $O\left(\dfrac{L}{\mu}\log\dfrac{1}{\bar{\epsilon}} + \dfrac{\sigma_{\text{tot}}^2}{\mu\epsilon}\right)$ with $\bar{\epsilon} = O\left(\dfrac{\sigma_{\text{tot}}^2}{\mu}\right)$ |
| N-SAGA [14] | $\epsilon_0 = O\left(\dfrac{\sigma_p^2}{\mu}\right)$ | $O\left(\left(n + \dfrac{L}{\mu}\right)\log\dfrac{1}{\epsilon}\right)$ with $\epsilon > \epsilon_0$ |
| S-MISO | 0 | $O\left(\left(n + \dfrac{L}{\mu}\right)\log\dfrac{1}{\bar{\epsilon}} + \dfrac{\sigma_p^2}{\mu\epsilon}\right)$ with $\bar{\epsilon} = O\left(\dfrac{\sigma_p^2}{\mu}\right)$ |

larger datasets [19, 26, 30]. Injecting noise in data is also useful to hide gradient information for privacy-aware learning [10].

Despite its importance, the optimization problem (2) has been littled studied and to the best of our knowledge, no dedicated optimization method that is able to exploit the problem structure has been developed so far. A natural way to optimize this objective when $h = 0$ is indeed SGD, but ignoring the finite-sum structure leads to gradient estimates with high variance and slow convergence. The goal of this paper is to introduce an algorithm for strongly convex objectives, called *stochastic MISO*, which exploits the underlying finite sum using variance reduction. Our method achieves a faster convergence rate than SGD, by removing the dependence on the gradient variance due to sampling the data points $i$ in $\{1, \ldots, n\}$; the dependence remains only for the variance due to random perturbations $\rho$.

To the best of our knowledge, our method is the first algorithm that interpolates naturally between incremental methods for finite sums (when there are no perturbations) and the stochastic approximation setting (when $n = 1$), while being able to efficiently tackle the hybrid case.

**Related work.** Many optimization methods dedicated to the finite-sum problem (*e.g.*, [15, 29]) have been motivated by the fact that their updates can be interpreted as SGD steps with unbiased estimates of the full gradient, but with a variance that decreases as the algorithm approaches the optimum [15]; on the other hand, vanilla SGD requires decreasing step-sizes to achieve this reduction of variance, thereby slowing down convergence. Our work aims at extending these techniques to the case where each function in the finite sum can only be accessed via a first-order stochastic oracle.

Most related to our work, recent methods that use data clustering to accelerate variance reduction techniques [3, 14] can be seen as tackling a special case of (2), where the expectations in $f_i$ are replaced by empirical averages over points in a cluster. While N-SAGA [14] was originally not designed for the stochastic context we consider, we remark that their method can be applied to (2). Their algorithm is however asymptotically biased and does not converge to the optimum. On the other hand, ClusterSVRG [3] is not biased, but does not support infinite datasets. The method proposed in [1] uses variance reduction in a setting where gradients are computed approximately, but the algorithm computes a full gradient at every pass, which is not available in our stochastic setting.

**Paper organization.** In Section 2, we present our algorithm for smooth objectives, and we analyze its convergence in Section 3. For space limitation reasons, we present an extension to composite objectives and non-uniform sampling in Appendix A. Section 4 is devoted to empirical results.

## 2 The Stochastic MISO Algorithm for Smooth Objectives

In this section, we introduce the *stochastic MISO* approach for smooth objectives ($h = 0$), which relies on the following assumptions:

- (A1) **global strong convexity**: $f$ is $\mu$-strongly convex;
- (A2) **smoothness**: $\tilde{f}_i(\cdot, \rho)$ is $L$-smooth for all $i$ and $\rho$ (*i.e.*, with $L$-Lipschitz gradients).

Table 2: Estimated ratio $\sigma_{\text{tot}}^2/\sigma_p^2$, which corresponds to the expected acceleration of S-MISO over SGD. These numbers are based on feature vectors variance, which is closely related to the gradient variance when learning a linear model. ResNet-50 denotes a 50 layer network [12] pre-trained on the ImageNet dataset. For image transformations, the numbers are empirically evaluated from 100 different images, with 100 random perturbations for each image. $R_{\text{tot}}^2$ (respectively, $R_{\text{cluster}}^2$) denotes the average squared distance between pairs of points in the dataset (respectively, in a given cluster), following [14]. The settings for unsupervised CKN and Scattering are described in Section 4. More details are given in the main text.

| Type of perturbation | Application case | Estimated ratio $\sigma_{\text{tot}}^2/\sigma_p^2$ | |
|---|---|---|---|
| Direct perturbation of linear model features | Data clustering as in [3, 14] | $\approx$ | $R_{\text{tot}}^2/R_{\text{cluster}}^2$ |
| | Additive Gaussian noise $\mathcal{N}(0, \alpha^2 I)$ | $\approx$ | $1 + 1/\alpha^2$ |
| | Dropout with probability $\delta$ | $\approx$ | $1 + 1/\delta$ |
| | Feature rescaling by $s$ in $\mathcal{U}(1-w, 1+w)$ | $\approx$ | $1 + 3/w^2$ |
| Random image transformations | ResNet-50 [12], color perturbation | 21.9 | |
| | ResNet-50 [12], rescaling + crop | 13.6 | |
| | Unsupervised CKN [22], rescaling + crop | 9.6 | |
| | Scattering [6], gamma correction | 9.8 | |

Note that these assumptions are relaxed in Appendix A by supporting composite objectives and by exploiting different smoothness parameters $L_i$ on each example, a setting where non-uniform sampling of the training points is typically helpful to accelerate convergence (*e.g.*, [35]).

**Complexity results.** We now introduce the following quantity, which is essential in our analysis:

$$\sigma_p^2 := \frac{1}{n}\sum_{i=1}^{n}\sigma_i^2, \;\; \text{with} \;\; \sigma_i^2 := \mathbb{E}_\rho\left[\|\nabla\tilde{f}_i(x^*, \rho) - \nabla f_i(x^*)\|^2\right],$$

where $x^*$ is the (unique) minimizer of $f$. The quantity $\sigma_p^2$ represents the part of the variance of the gradients at the optimum that is due to the perturbations $\rho$. In contrast, another quantity of interest is the total variance $\sigma_{\text{tot}}^2$, which also includes the randomness in the choice of the index $i$, defined as

$$\sigma_{\text{tot}}^2 = \mathbb{E}_{i,\rho}[\|\nabla\tilde{f}_i(x^*,\rho)\|^2] = \sigma_p^2 + \mathbb{E}_i[\|\nabla f_i(x^*)\|^2] \qquad \text{(note that } \nabla f(x^*) = 0).$$

The relation between $\sigma_{\text{tot}}^2$ and $\sigma_p^2$ is obtained by simple algebraic manipulations.

The goal of our paper is to exploit the potential imbalance $\sigma_p^2 \ll \sigma_{\text{tot}}^2$, occurring when perturbations on input data are small compared to the sampling noise. The assumption is reasonable: given a data point, selecting a different one should lead to larger variation than a simple perturbation. From a theoretical point of view, the approach we propose achieves the iteration complexity presented in Table 1, see also Appendix D and [4, 5, 24] for the complexity analysis of SGD. The gain over SGD is of order $\sigma_{\text{tot}}^2/\sigma_p^2$, which is also observed in our experiments in Section 4. We also compare against the method N-SAGA; its convergence rate is similar to ours but suffers from a non-zero asymptotic error.

**Motivation from application cases.** One clear framework of application is the data clustering scenario already investigated in [3, 14]. Nevertheless, we will focus on less-studied data augmentation settings that lead instead to true stochastic formulations such as (2). First, we consider learning a linear model when adding simple direct manipulations of feature vectors, via rescaling (multiplying each entry vector by a random scalar), Dropout, or additive Gaussian noise, in order to improve the generalization error [33] or to get more stable estimators [23]. In Table 2, we present the potential gain over SGD in these scenarios. To do that, we study the variance of perturbations applied to a feature vector $\xi$. Indeed, the gradient of the loss is proportional to $\xi$, which allows us to obtain good estimates of the ratio $\sigma_{\text{tot}}^2/\sigma_p^2$, as we observed in our empirical study of Dropout presented in Section 4. Whereas some perturbations are friendly for our method such as feature rescaling (a rescaling window of $[0.9, 1.1]$ yields for instance a huge gain factor of 300), a large Dropout rate would lead to less impressive acceleration (*e.g.*, a Dropout with $\delta = 0.5$ simply yields a factor 2).

Second, we also consider more interesting domain-driven data perturbations such as classical image transformations considered in computer vision [26, 36] including image cropping, rescaling, brightness, contrast, hue, and saturation changes. These transformations may be used to train a linear

---

**Algorithm 1** S-MISO for smooth objectives

---

**Input:** step-size sequence $(\alpha_t)_{t \geq 1}$;
initialize $x_0 = \frac{1}{n} \sum_i z_i^0$ for some $(z_i^0)_{i=1,\ldots,n}$;
**for** $t = 1, \ldots$ **do**

   Sample an index $i_t$ uniformly at random, a perturbation $\rho_t$, and update

$$z_i^t = \begin{cases} (1 - \alpha_t)z_i^{t-1} + \alpha_t(x_{t-1} - \frac{1}{\mu}\nabla \tilde{f}_{i_t}(x_{t-1}, \rho_t)), & \text{if } i = i_t \\ z_i^{t-1}, & \text{otherwise.} \end{cases} \quad (3)$$

$$x_t = \frac{1}{n}\sum_{i=1}^n z_i^t = x_{t-1} + \frac{1}{n}(z_{i_t}^t - z_{i_t}^{t-1}). \quad (4)$$

  **end for**

---

classifier on top of an unsupervised multilayer image model such as unsupervised CKNs [22] or the scattering transform [6]. It may also be used for retraining the last layer of a pre-trained deep neural network: given a new task unseen during the full network training and given limited amount of training data, data augmentation may be indeed crucial to obtain good prediction and S-MISO can help accelerate learning in this setting. These scenarios are also studied in Table 2, where the experiment with ResNet-50 involving random cropping and rescaling produces $224 \times 224$ images from $256 \times 256$ ones. For these scenarios with realistic perturbations, the potential gain varies from 10 to 20.

**Description of stochastic MISO.**  We are now in shape to present our method, described in Algorithm 1. Without perturbations and with a constant step-size, the algorithm resembles the MISO/Finito algorithms [9, 18, 21], which may be seen as primal variants of SDCA [28, 29]. Specifically, MISO is not able to deal with our stochastic objective (2), but it may address the deterministic finite-sum problem (1). It is part of a larger body of optimization methods that iteratively build a *model* of the objective function, typically a lower or upper bound on the objective that is easier to optimize; for instance, this strategy is commonly adopted in bundle methods [13, 25].

More precisely, MISO assumes that each $f_i$ is strongly convex and builds a model using lower bounds $D_t(x) = \frac{1}{n}\sum_{i=1}^n d_i^t(x)$, where each $d_i^t$ is a quadratic lower bound on $f_i$ of the form

$$d_i^t(x) = c_{i,1}^t + \frac{\mu}{2}\|x - z_i^t\|^2 = c_{i,2}^t - \mu\langle x, z_i^t \rangle + \frac{\mu}{2}\|x\|^2. \quad (5)$$

These lower bounds are updated during the algorithm using strong convexity lower bounds at $x_{t-1}$ of the form $l_i^t(x) = f_i(x_{t-1}) + \langle \nabla f_i(x_{t-1}), x - x_{t-1} \rangle + \frac{\mu}{2}\|x - x_{t-1}\|^2 \leq f_i(x)$:

$$d_i^t(x) = \begin{cases} (1 - \alpha_t)d_i^{t-1}(x) + \alpha_t l_i^t(x), & \text{if } i = i_t \\ d_i^{t-1}(x), & \text{otherwise,} \end{cases} \quad (6)$$

which corresponds to an update of the quantity $z_i^t$:

$$z_i^t = \begin{cases} (1 - \alpha_t)z_i^{t-1} + \alpha_t(x_{t-1} - \frac{1}{\mu}\nabla f_{i_t}(x_{t-1})), & \text{if } i = i_t \\ z_i^{t-1}, & \text{otherwise.} \end{cases}$$

The next iterate is then computed as $x_t = \arg\min_x D_t(x)$, which is equivalent to (4). The original MISO/Finito algorithms use $\alpha_t = 1$ under a "big data" condition on the sample size $n$ [9, 21], while the theory was later extended in [18] to relax this condition by supporting smaller constant steps $\alpha_t = \alpha$, leading to an algorithm that may be interpreted as a primal variant of SDCA (see [28]).

Note that when $f_i$ is an expectation, it is hard to obtain such lower bounds since the gradient $\nabla f_i(x_{t-1})$ is not available in general. For this reason, we have introduced S-MISO, which can exploit *approximate* lower bounds to each $f_i$ using gradient estimates, by letting the step-sizes $\alpha_t$ decrease appropriately as commonly done in stochastic approximation. This leads to update (3).

Separately, SDCA [29] considers the Fenchel conjugates of $f_i$, defined by $f_i^*(y) = \sup_x x^\top y - f_i(x)$. When $f_i$ is an expectation, $f_i^*$ is not available in closed form in general, nor are its gradients, and in fact exploiting stochastic gradient estimates is difficult in the duality framework. In contrast, [28] gives an analysis of SDCA in the primal, aka. "without duality", for smooth finite sums, and our work extends this line of reasoning to the stochastic approximation and composite settings.

**Relationship with SGD in the smooth case.** The link between S-MISO in the non-composite setting and SGD can be seen by rewriting the update (4) as

$$x_t = x_{t-1} + \frac{1}{n}(z_{i_t}^t - z_{i_t}^{t-1}) = x_{t-1} + \frac{\alpha_t}{n}v_t,$$

where
$$v_t := x_{t-1} - \frac{1}{\mu}\nabla\tilde{f}_{i_t}(x_{t-1}, \rho_t) - z_{i_t}^{t-1}. \tag{7}$$

Note that $\mathbb{E}[v_t|\mathcal{F}_{t-1}] = -\frac{1}{\mu}\nabla f(x_{t-1})$, where $\mathcal{F}_{t-1}$ contains all information up to iteration $t$; hence, the algorithm can be seen as an instance of the stochastic gradient method with unbiased gradients, which was a key motivation in SVRG [15] and later in other variance reduction algorithms [8, 28]. It is also worth noting that in the absence of a finite-sum structure ($n=1$), we have $z_{i_t}^{t-1} = x_{t-1}$; hence our method becomes identical to SGD, up to a redefinition of step-sizes. In the composite case (see Appendix A), our approach yields a new algorithm that resembles regularized dual averaging [34].

**Memory requirements and handling of sparse datasets.** The algorithm requires storing the vectors $(z_i^t)_{i=1,\ldots,n}$, which takes the same amount of memory as the original dataset and which is therefore a reasonable requirement in many practical cases. In the case of sparse datasets, it is fair to assume that random perturbations applied to input data preserve the sparsity patterns of the original vectors, as is the case, *e.g.*, when applying Dropout to text documents described with bag-of-words representations [33]. If we further assume the typical setting where the $\mu$-strong convexity comes from an $\ell_2$ regularizer: $\tilde{f}_i(x, \rho) = \phi_i(x^\top \xi_i^\rho) + (\mu/2)\|x\|^2$, where $\xi_i^\rho$ is the (sparse) perturbed example and $\phi_i$ encodes the loss, then the update (3) can be written as

$$z_i^t = \begin{cases} (1 - \alpha_t)z_i^{t-1} - \frac{\alpha_t}{\mu}\phi_i'(x_{t-1}^\top \xi_i^{\rho_t})\xi_i^{\rho_t}, & \text{if } i = i_t \\ z_i^{t-1}, & \text{otherwise,} \end{cases}$$

which shows that for every index $i$, the vector $z_i^t$ preserves the same sparsity pattern as the examples $\xi_i^\rho$ throughout the algorithm (assuming the initialization $z_i^0 = 0$), making the update (3) efficient. The update (4) has the same cost since $v_t = z_{i_t}^t - z_{i_t}^{t-1}$ is also sparse.

**Limitations and alternative approaches.** Since our algorithm is uniformly better than SGD in terms of iteration complexity, its main limitation is in terms of memory storage when the dataset cannot fit into memory (remember that the memory cost of S-MISO is the same as the input dataset). In these huge-scale settings, SGD should be preferred; this holds true in fact for all incremental methods when one cannot afford to perform more than one (or very few) passes over the data. Our paper focuses instead on non-huge datasets, which are those benefiting most from data augmentation.

We note that a different approach to variance reduction like SVRG [15] is able to trade off storage requirements for additional full gradient computations, which would be desirable in some situations. However, we were not able to obtain any decreasing step-size strategy that works for these methods, both in theory and practice, leaving us with constant step-size approaches as in [1, 14] that either maintain a non-zero asymptotic error, or require dynamically reducing the variance of gradient estimates. One possible way to explain this difficulty is that SVRG and SAGA [8] "forget" past gradients for a given example $i$, while S-MISO averages them in (3), which seems to be a technical key to make it suitable to stochastic approximation. Nevertheless, the question of whether it is possible to trade-off storage with computation in a setting like ours is open and of utmost interest.

## 3 Convergence Analysis of S-MISO

We now study the convergence properties of the S-MISO algorithm. For space limitation reasons, all proofs are provided in Appendix B. We start by defining the problem-dependent quantities $z_i^* := x^* - \frac{1}{\mu}\nabla f_i(x^*)$, and then introduce the Lyapunov function

$$C_t = \frac{1}{2}\|x_t - x^*\|^2 + \frac{\alpha_t}{n^2}\sum_{i=1}^{n}\|z_i^t - z_i^*\|^2. \tag{8}$$

Proposition 1 gives a recursion on $C_t$, obtained by upper-bounding separately its two terms, and finding coefficients to cancel out other appearing quantities when relating $C_t$ to $C_{t-1}$. To this end, we borrow elements of the convergence proof of SDCA without duality [28]; our technical contribution is to extend their result to the stochastic approximation and composite (see Appendix A) cases.

**Proposition 1** (Recursion on $C_t$)**.** *If $(\alpha_t)_{t \geq 1}$ is a positive and non-increasing sequence satisfying*

$$\alpha_1 \leq \min\left\{\frac{1}{2}, \frac{n}{2(2\kappa - 1)}\right\},\tag{9}$$

*with $\kappa = L/\mu$, then $C_t$ obeys the recursion*

$$\mathbb{E}[C_t] \leq \left(1 - \frac{\alpha_t}{n}\right)\mathbb{E}[C_{t-1}] + 2\left(\frac{\alpha_t}{n}\right)^2\frac{\sigma_p^2}{\mu^2}.\tag{10}$$

We now state the main convergence result, which provides the expected rate $O(1/t)$ on $C_t$ based on decreasing step-sizes, similar to [5] for SGD. Note that convergence of objective function values is directly related to that of the Lyapunov function $C_t$ via smoothness:

$$\mathbb{E}[f(x_t) - f(x^*)] \leq \frac{L}{2}\mathbb{E}\left[\|x_t - x^*\|^2\right] \leq L\,\mathbb{E}[C_t].\tag{11}$$

**Theorem 2** (Convergence of Lyapunov function)**.** *Let the sequence of step-sizes $(\alpha_t)_{t \geq 1}$ be defined by $\alpha_t = \frac{2n}{\gamma + t}$ with $\gamma \geq 0$ such that $\alpha_1$ satisfies (9). For all $t \geq 0$, it holds that*

$$\mathbb{E}[C_t] \leq \frac{\nu}{\gamma + t + 1} \quad where \quad \nu := \max\left\{\frac{8\sigma_p^2}{\mu^2}, (\gamma + 1)C_0\right\}.\tag{12}$$

**Choice of step-sizes in practice.** Naturally, we would like $\nu$ to be small, in particular independent of the initial condition $C_0$ and equal to the first term in the definition (12). We would like the dependence on $C_0$ to vanish at a faster rate than $O(1/t)$, as it is the case in variance reduction algorithms on finite sums. As advised in [5] in the context of SGD, we can initially run the algorithm with a constant step-size $\bar{\alpha}$ and exploit this linear convergence regime until we reach the level of noise given by $\sigma_p$, and then start decaying the step-size. It is easy to see that by using a constant step-size $\bar{\alpha}$, $C_t$ converges near a value $\bar{C} := 2\bar{\alpha}\sigma_p^2/n\mu^2$. Indeed, Eq. (10) with $\alpha_t = \bar{\alpha}$ yields

$$\mathbb{E}[C_t - \bar{C}] \leq \left(1 - \frac{\bar{\alpha}}{n}\right)\mathbb{E}[C_{t-1} - \bar{C}].$$

Thus, we can reach a precision $C_0'$ with $\mathbb{E}[C_0'] \leq \bar{\epsilon} := 2\bar{C}$ in $O(\frac{n}{\bar{\alpha}}\log C_0/\bar{\epsilon})$ iterations. Then, if we start decaying step-sizes as in Theorem 2 with $\gamma$ large enough so that $\alpha_1 = \bar{\alpha}$, we have

$$(\gamma + 1)\mathbb{E}[C_0'] \leq (\gamma + 1)\bar{\epsilon} = 8\sigma_p^2/\mu^2,$$

making both terms in (12) smaller than or equal to $\nu = 8\sigma_p^2/\mu^2$. Considering these two phases, with an initial step-size $\bar{\alpha}$ given by (9), the final work complexity for reaching $\mathbb{E}[\|x_t - x^*\|^2] \leq \epsilon$ is

$$O\left(\left(n + \frac{L}{\mu}\right)\log\frac{C_0}{\bar{\epsilon}}\right) + O\left(\frac{\sigma_p^2}{\mu^2\epsilon}\right).\tag{13}$$

We can then use (11) in order to obtain the complexity for reaching $\mathbb{E}[f(x_t) - f(x^*)] \leq \epsilon$. Note that following this step-size strategy was found to be very effective in practice (see Section 4).

**Acceleration by iterate averaging.** When one is interested in the convergence in function values, the complexity (13) combined with (11) yields $O(L\sigma_p^2/\mu^2\epsilon)$, which can be problematic for ill-conditioned problems (large condition number $L/\mu$). The following theorem presents an iterate averaging scheme which brings the complexity term down to $O(\sigma_p^2/\mu\epsilon)$, which appeared in Table 1.

**Theorem 3** (Convergence under iterate averaging)**.** *Let the step-size sequence $(\alpha_t)_{t \geq 1}$ be defined by*

$$\alpha_t = \frac{2n}{\gamma + t} \quad for\ \gamma \geq 1\ s.t.\ \alpha_1 \leq \min\left\{\frac{1}{2}, \frac{n}{4(2\kappa - 1)}\right\}.$$

*We have*

$$\mathbb{E}[f(\bar{x}_T) - f(x^*)] \leq \frac{2\mu\gamma(\gamma - 1)C_0}{T(2\gamma + T - 1)} + \frac{16\sigma_p^2}{\mu(2\gamma + T - 1)},$$

*where*

$$\bar{x}_T := \frac{2}{T(2\gamma + T - 1)}\sum_{t=0}^{T-1}(\gamma + t)x_t.$$

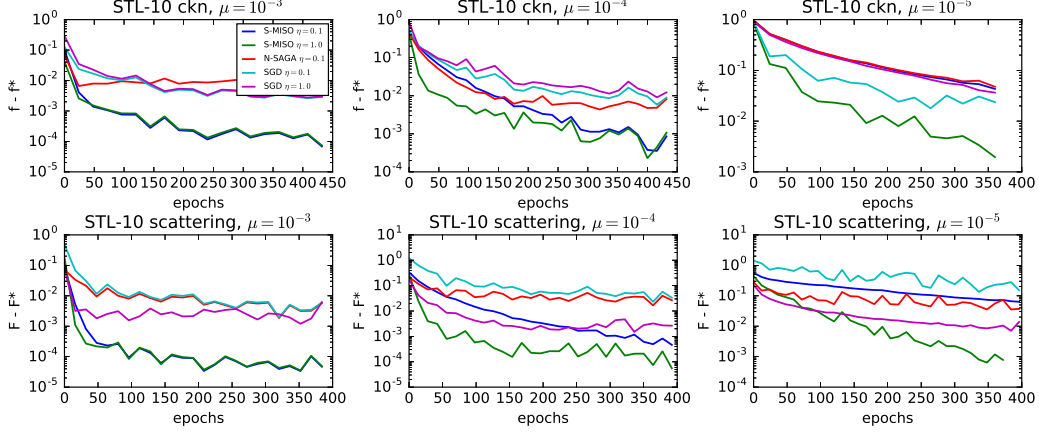

Figure 1: Impact of conditioning for data augmentation on STL-10 (controlled by $\mu$, where $\mu = 10^{-4}$ gives the best accuracy). Values of the loss are shown on a **logarithmic scale** (1 unit = factor 10). $\eta = 0.1$ satisfies the theory for all methods, and we include curves for larger step-sizes $\eta = 1$. We omit N-SAGA for $\eta = 1$ because it remains far from the optimum. For the scattering representation, the problem we study is $\ell_1$-regularized, and we use the composite algorithm of Appendix A.

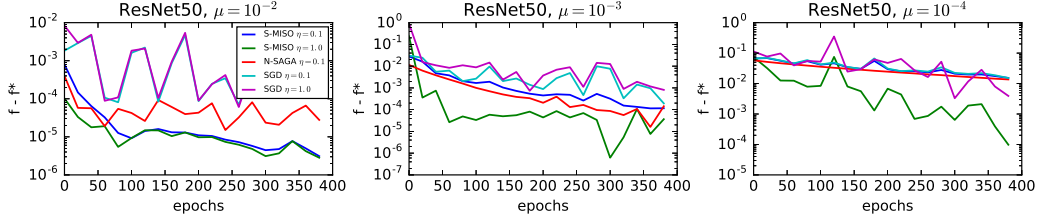

Figure 2: Re-training of the last layer of a pre-trained ResNet 50 model, on a small dataset with random color perturbations (for different values of $\mu$).

The proof uses a similar telescoping sum technique to [16]. Note that if $T \gg \gamma$, the first term, which depends on the initial condition $C_0$, decays as $1/T^2$ and is thus dominated by the second term. Moreover, if we start averaging after an initial phase with constant step-size $\bar{\alpha}$, we can consider $C_0 \approx 4\bar{\alpha}\sigma_p^2/n\mu^2$. In the ill-conditioned regime, taking $\bar{\alpha} = \alpha_1 = 2n/(\gamma + 1)$ as large as allowed by (9), we have $\gamma$ of the order of $\kappa = L/\mu \gg 1$. The full convergence rate then becomes

$$\mathbb{E}[f(\bar{x}_T) - f(x^*)] \leq O\left(\frac{\sigma_p^2}{\mu(\gamma + T)}\left(1 + \frac{\gamma}{T}\right)\right).$$

When $T$ is large enough compared to $\gamma$, this becomes $O(\sigma_p^2/\mu T)$, leading to a complexity $O(\sigma_p^2/\mu\epsilon)$.

## 4 Experiments

We present experiments comparing S-MISO with SGD and N-SAGA [14] on four different scenarios, in order to demonstrate the wide applicability of our method: we consider an image classification dataset with two different image representations and random transformations, and two classification tasks with Dropout regularization, one on genetic data, and one on (sparse) text data. Figures 1 and 3 show the curves for an estimate of the training objective using 5 sampled perturbations per example. The plots are shown on a logarithmic scale, and the values are compared to the best value obtained among the different methods in 500 epochs. The strong convexity constant $\mu$ is the regularization parameter. For all methods, we consider step-sizes supported by the theory as well as larger step-sizes that may work better in practice. Our C++/Cython implementation of all methods considered in this section is available at `https://github.com/albietz/stochs`.

**Choices of step-sizes.** For both S-MISO and SGD, we use the step-size strategy mentioned in Section 3 and advised by [5], which we have found to be most effective among many heuristics

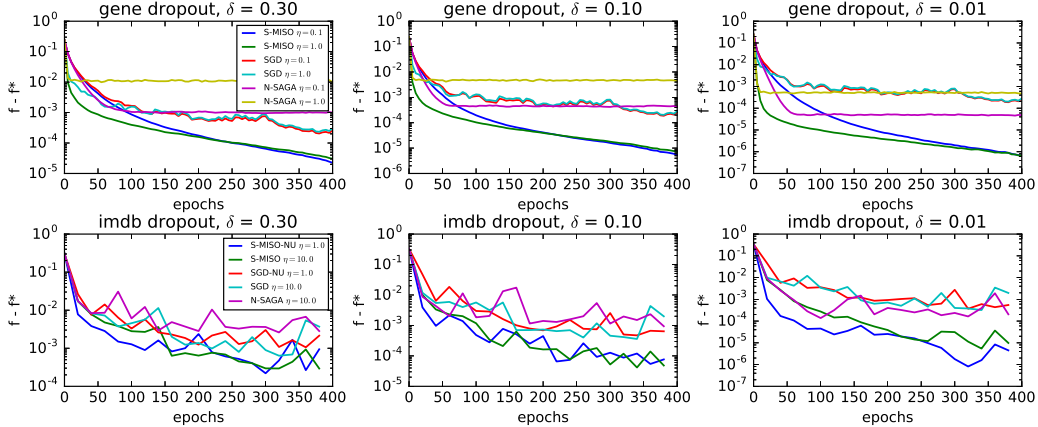

Figure 3: Impact of perturbations controlled by the Dropout rate $\delta$. The gene data is $\ell_2$-normalized; hence, we consider similar step-sizes as Figure 1. The IMDB dataset is highly heterogeneous; thus, we also include non-uniform (NU) sampling variants of Appendix A. For uniform sampling, theoretical step-sizes perform poorly for all methods; thus, we show a larger tuned step-size $\eta = 10$.

we have tried: we initially keep the step-size constant (controlled by a factor $\eta \leq 1$ in the figures) for 2 epochs, and then start decaying as $\alpha_t = C/(\gamma + t)$, where $C = 2n$ for S-MISO, $C = 2/\mu$ for SGD, and $\gamma$ is chosen large enough to match the previous constant step-size. For N-SAGA, we maintain a constant step-size throughout the optimization, as suggested in the original paper [14]. The factor $\eta$ shown in the figures is such that $\eta = 1$ corresponds to an initial step-size $n\mu/(L - \mu)$ for S-MISO (from (19) in the uniform case) and $1/L$ for SGD and N-SAGA (with $\bar{L}$ instead of $L$ in the non-uniform case when using the variant of Appendix A).

**Image classification with "data augmentation".** The success of deep neural networks is often limited by the availability of large amounts of labeled images. When there are many unlabeled images but few labeled ones, a common approach is to train a linear classifier on top of a deep network learned in an unsupervised manner, or pre-trained on a different task (*e.g.*, on the ImageNet dataset). We follow this approach on the STL-10 dataset [7], which contains 5K training images from 10 classes and 100K unlabeled images, using a 2-layer unsupervised convolutional kernel network [22], giving representations of dimension $9\,216$. The perturbation consists of randomly cropping and scaling the input images. We use the squared hinge loss in a one-versus-all setting. The vector representations are $\ell_2$-normalized such that we may use the upper bound $L = 1 + \mu$ for the smoothness constant. We also present results on the same dataset using a scattering representation [6] of dimension $21\,696$, with random gamma corrections (raising all pixels to the power $\gamma$, where $\gamma$ is chosen randomly around 1). For this representation, we add an $\ell_1$ regularization term and use the composite variant of S-MISO presented in Appendix A.

Figure 1 shows convergence results on one training fold (500 images), for different values of $\mu$, allowing us to study the behavior of the algorithms for different condition numbers. The low variance induced by data transformations allows S-MISO to reach suboptimality that is orders of magnitude smaller than SGD after the same number of epochs. Note that one unit on these plots corresponds to one order of magnitude in the logarithmic scale. N-SAGA initially reaches a smaller suboptimality than SGD, but quickly gets stuck due to the bias in the algorithm, as predicted by the theory [14], while S-MISO and SGD continue to converge to the optimum thanks to the decreasing step-sizes. The best validation accuracy for both representations is obtained for $\mu \approx 10^{-4}$ (middle column), and we observed relative gains of up to 1% from using data augmentation. We computed empirical variances of the image representations for these two strategies, which are closely related to the variance in gradient estimates, and observed these transformations to account for about 10% of the total variance.

Figure 2 shows convergence results when training the last layer of a 50-layer Residual network [12] that has been pre-trained on ImageNet. Here, we consider the common scenario of leveraging a deep model trained on a large dataset as a feature extractor in order to learn a new classifier on a different small dataset, where it would be difficult to train such a model from scratch. To simulate this setting, we consider a binary classification task on a small dataset of 100 images of size 256x256 taken from the ImageNet Large Scale Visual Recognition Challenge (ILSVRC) 2012, which we crop to

224x224 before performing random adjustments to brightness, saturation, hue and contrast. As in the STL-10 experiments, the gains of S-MISO over other methods are of about one order of magnitude in suboptimality, as predicted by Table 2.

**Dropout on gene expression data.** We trained a binary logistic regression model on the breast cancer dataset of [31], with different Dropout rates $\delta$, *i.e.*, where at every iteration, each coordinate $\xi_j$ of a feature vector $\xi$ is set to zero independently with probability $\delta$ and to $\xi_j/(1-\delta)$ otherwise. The dataset consists of 295 vectors of dimension 8 141 of gene expression data, which we normalize in $\ell_2$ norm. Figure 3 (top) compares S-MISO with SGD and N-SAGA for three values of $\delta$, as a way to control the variance of the perturbations. We include a Dropout rate of 0.01 to illustrate the impact of $\delta$ on the algorithms and study the influence of the perturbation variance $\sigma_p^2$, even though this value of $\delta$ is less relevant for the task. The plots show very clearly how the variance induced by the perturbations affects the convergence of S-MISO, giving suboptimality values that may be orders of magnitude smaller than SGD. This behavior is consistent with the theoretical convergence rate established in Section 3 and shows that the practice matches the theory.

**Dropout on movie review sentiment analysis data.** We trained a binary classifier with a squared hinge loss on the IMDB dataset [20] with different Dropout rates $\delta$. We use the labeled part of the IMDB dataset, which consists of 25K training and 250K testing movie reviews, represented as 89 527-dimensional sparse bag-of-words vectors. In contrast to the previous experiments, we do not normalize the representations, which have great variability in their norms, in particular, the maximum Lipschitz constant across training points is roughly 100 times larger than the average one. Figure 3 (bottom) compares non-uniform sampling versions of S-MISO (see Appendix A) and SGD (see Appendix D) with their uniform sampling counterparts as well as N-SAGA. Note that we use a large step-size $\eta = 10$ for the uniform sampling algorithms, since $\eta = 1$ was significantly slower for all methods, likely due to outliers in the dataset. In contrast, the non-uniform sampling algorithms required no tuning and just use $\eta = 1$. The curves clearly show that S-MISO-NU has a much faster convergence in the initial phase, thanks to the larger step-size allowed by non-uniform sampling, and later converges similarly to S-MISO, *i.e.*, at a much faster rate than SGD when the perturbations are small. The value of $\mu$ used in the experiments was chosen by cross-validation, and the use of Dropout gave improvements in test accuracy from 88.51% with no dropout to $88.68 \pm 0.03\%$ with $\delta = 0.1$ and $88.86 \pm 0.11\%$ with $\delta = 0.3$ (based on 10 different runs of S-MISO-NU after 400 epochs).

Finally, we also study the effect of the iterate averaging scheme of Theorem 3 in Appendix E.

### Acknowledgements

This work was supported by a grant from ANR (MACARON project under grant number ANR-14-CE23-0003-01), by the ERC grant number 714381 (SOLARIS project), and by the MSR-Inria joint center.

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
