[Supplementary Material · nips_appendix.pdf]

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

**Algorithm 2** S-MISO for composite objectives, with non-uniform sampling.

---

**Input:** step-sizes $(\alpha_t)_{t \geq 1}$, sampling distribution $q$;
Initialize $x_0 = \text{prox}_{h/\mu}(\bar{z}_0)$ with $\bar{z}_0 = \frac{1}{n}\sum_i z_i^0$ for some $(z_i^0)_{i=1,\ldots,n}$ that satisfies (16);
**for** $t = 1, \ldots$ **do**
    Sample an index $i_t \sim q$, a perturbation $\rho_t$, and update (with $\alpha_t^i = \frac{\alpha_t}{q_i n}$):

$$z_i^t = \begin{cases} (1 - \alpha_t^i)z_i^{t-1} + \alpha_t^i(x_{t-1} - \frac{1}{\mu}\nabla \tilde{f}_{i_t}(x_{t-1}, \rho_t)), & \text{if } i = i_t \\ z_i^{t-1}, & \text{otherwise} \end{cases} \tag{14}$$

$$\bar{z}_t = \frac{1}{n}\sum_{i=1}^n z_i^t = \bar{z}_{t-1} + \frac{1}{n}(z_{i_t}^t - z_{i_t}^{t-1})$$

$$x_t = \text{prox}_{h/\mu}(\bar{z}_t). \tag{15}$$

**end for**

---

## A  Extension to Composite Objectives and Non-Uniform Sampling

In this section, we study extensions of S-MISO to different situations where our previous smoothness assumption (A2) is not suitable, either because of a non-smooth term $h$ in the objective or because it ignores additional useful knowledge about each $f_i$ such as the norm of each example.

In the presence of non-smooth regularizers such as the $\ell_1$-norm, the objective is no longer smooth, but we can leverage its composite structure by using proximal operators. To this end, we assume that one can easily compute the proximal operator of $h$, defined by

$$\text{prox}_h(z) := \arg\min_{x \in \mathbb{R}^p} \left\{ \frac{1}{2}\|x - z\|^2 + h(x) \right\}.$$

When the smoothness constants $L_i$ vary significantly across different examples (typically through the norm of the feature vectors), the uniform upper bound $L = L_{\max} = \max_i L_i$ can be restrictive. It has been noticed (see, *e.g.*, [27, 35]) that when the $L_i$ are known, one can achieve better convergence rates—typically depending on the average smoothness constant $\bar{L} = \frac{1}{n}\sum_i L_i$ rather than $L_{\max}$—by sampling examples in a non-uniform way. For that purpose, we now make the following assumptions:

- (A3) **strong convexity**: $\tilde{f}_i(\cdot, \rho)$ is $\mu$-strongly convex for all $i, \rho$;
- (A4) **smoothness**: $\tilde{f}_i(\cdot, \rho)$ is $L_i$-smooth for all $i, \rho$;

Note that our proof relies on a slightly stronger assumption (A3) than the global strong convexity assumption (A1) made above, which holds in the situation where strong convexity comes from an $\ell_2$ regularization term. In order to exploit the different smoothness constants, we allow the algorithm to sample indices $i$ non-uniformly, from any distribution $q$ such that $q_i \geq 0$ for all $i$ and $\sum_i q_i = 1$.

The extension of S-MISO to this setting is given in Algorithm 2. Note that the step-sizes vary depending on the example, with larger steps for examples that are sampled less frequently (typically "easier" examples with smaller $L_i$). Note that when $h = 0$, the update directions are unbiased estimates of the gradient: we have $\mathbb{E}[x_t - x_{t-1}|\mathcal{F}_{t-1}] = -\frac{\alpha_t}{n\mu}\nabla f(x_{t-1})$ as in the uniform case. However, in the composite case, the algorithm cannot be written in a proximal stochastic gradient form like Prox-SVRG [35] or SAGA [8].

**Relationship with RDA.**  When $n = 1$, our algorithm performs similar updates to Regularized Dual Averaging (RDA) [34] with strongly convex regularizers. In particular, if $\tilde{f}_1(x, \rho) = \phi(x^\top \xi(\rho)) + (\mu/2)\|x\|^2$, the updates are the same when taking $\alpha_t = 1/t$, since

$$\text{prox}_{h/\mu}(\bar{z}_t) = \arg\min_x \left\{ \langle -\mu\bar{z}_t, x \rangle + \frac{\mu}{2}\|x\|^2 + h(x) \right\},$$

and $-\mu\bar{z}_t$ is equal to the average of the gradients of the loss term up to $t$, which appears in the same way in the RDA updates [34, Section 2.2]. However, unlike RDA, our method supports arbitrary decreasing step-sizes, in particular keeping the step-size constant, which can lead to faster convergence in the initial iterations (see Section 3).

**Lower-bound model and convergence analysis.** Again, we can view the algorithm as iteratively updating approximate lower bounds on the objective $F$ of the form $D_t(x) = \frac{1}{n} \sum_i d_i^t(x) + h(x)$ analogously to (6), and minimizing the new $D_t$ in (15). Similar to MISO-Prox, we require that $d_i^0$ is initialized with a $\mu$-strongly convex quadratic such that $\tilde{f}_i(x, \tilde{\rho}_i) \geq d_i^0(x)$ with the random perturbation $\tilde{\rho}_i$. Given the form of $d_i^t$ in (5), it suffices to choose $z_i^0$ that satisfies

$$\tilde{f}_i(x, \tilde{\rho}_i) \geq \frac{\mu}{2} \|x - z_i^0\| + c, \tag{16}$$

for some constant $c$. In the common case of an $\ell_2$ regularizer with a non-negative loss, one can simply choose $z_i^0 = 0$ for all $i$, otherwise, $z_i^0$ can be obtained by considering a strong convexity lower bound on $\tilde{f}_i(\cdot, \tilde{\rho}_i)$. Our new analysis relies on the minimum $D_t(x_t)$ of the lower bounds $D_t$ through the following Lyapunov function:

$$C_t^q = F(x^*) - D_t(x_t) + \frac{\mu \alpha_t}{n^2} \sum_{i=1}^{n} \frac{1}{q_i n} \|z_i^t - z_i^*\|^2. \tag{17}$$

The convergence of the iterates $x_t$ is controlled by the convergence in $C_t^q$ thanks to the next lemma:

**Lemma 4** (Bound on the iterates). *For all $t$, we have*

$$\frac{\mu}{2} \mathbb{E}[\|x_t - x^*\|^2] \leq \mathbb{E}[F(x^*) - D_t(x_t)]. \tag{18}$$

The following proposition gives a recursion on $C_t^q$ similar to Proposition 1.

**Proposition 5** (Recursion on $C_t^q$). *If $(\alpha_t)_{t \geq 1}$ is a positive and non-increasing sequence of step-sizes satisfying*

$$\alpha_1 \leq \min \left\{ \frac{nq_{\min}}{2}, \frac{n\mu}{4L_q} \right\}, \tag{19}$$

*with $q_{\min} = \min_i q_i$ and $L_q = \max_i \frac{L_i - \mu}{q_i n}$, then $C_t^q$ obeys the recursion*

$$\mathbb{E}[C_t^q] \leq \left( 1 - \frac{\alpha_t}{n} \right) \mathbb{E}[C_{t-1}^q] + 2 \left( \frac{\alpha_t}{n} \right)^2 \frac{\sigma_q^2}{\mu}, \tag{20}$$

*with $\sigma_q^2 = \frac{1}{n} \sum_i \frac{\sigma_i^2}{q_i n}$.*

Note that if we consider the quantity $\mathbb{E}[C_t^q/\mu]$, which is an upper bound on $\frac{1}{2} \mathbb{E}[\|x_t - x^*\|^2]$ by Lemma 4, we have the same recursion as (10), and thus can apply Theorem 2 with the new condition (19). If we choose

$$q_i = \frac{1}{2n} + \frac{L_i - \mu}{2 \sum_i (L_i - \mu)}, \tag{21}$$

we have $q_{\min} \geq 1/2n$ and $L_q \leq 2(\bar{L} - \mu)$, where $\bar{L} = \frac{1}{n} \sum_i L_i$. Then, taking $\alpha_1 = \min(1/4, n\mu/8(\bar{L} - \mu))$ satisfies (19), and using similar arguments to Section 3, the complexity for reaching $\mathbb{E}[\|x_t - x^*\|^2] \leq \epsilon$ is

$$O \left( \left( n + \frac{\bar{L}}{\mu} \right) \log \frac{C_0^q}{\bar{\epsilon}} \right) + O \left( \frac{\sigma_q^2}{\mu^2 \epsilon} \right),$$

where $\bar{\epsilon} = 4\bar{\alpha} \sigma_q^2 / n\mu$, and $\bar{\alpha}$ is the initial constant step-size. For the complexity in function sub-optimality, the second term becomes $O(\sigma_q^2/\mu\epsilon)$ by using the same averaging scheme presented in Theorem 3 and adapting the proof. Note that with our choice of $q$, we have $\sigma_q^2 \leq \frac{2}{n} \sum_i \sigma_i^2 = 2\bar{\sigma}_p^2$, for general perturbations, where $\bar{\sigma}_p^2 = \frac{1}{n} \sum_i \sigma_i^2$ is the variance in the uniform case. Additionally, it is often reasonable to assume that the variance from perturbations increases with the norm of examples, for instance Dropout perturbations get larger when coordinates have larger magnitudes. Based on this observation, if we make the assumption that $\sigma_i^2 \propto L_i - \mu$, that is $\sigma_i^2 = \bar{\sigma}_p^2 \frac{L_i - \mu}{\bar{L} - \mu}$, then for both $q_i = 1/n$ (uniform case) and $q_i = (L_i - \mu)/n(\bar{L} - \mu)$, we have $\sigma_q^2 = \bar{\sigma}_p^2$, and thus we have $\sigma_q^2 \leq \bar{\sigma}_p^2$ for the choice of $q$ given in (21), since $\sigma_q^2$ is convex in $q$. Thus, we can expect that the $O(1/t)$ convergence phase behaves similarly or better than for uniform sampling, which is confirmed by our experiments (see Section 4).

# B Proofs for the Smooth Case (Section 3)

## B.1 Proof of Proposition 1 (Recursion on Lyapunov function $C_t$)

We begin by stating the following lemma, which extends a key result of variance reduction methods (see, *e.g.*, [15]) to the situation considered in this paper, where one only has access to noisy estimates of the gradients of each $f_i$.

**Lemma B.1.** *Let $i$ be uniformly distributed in $\{1, \ldots, n\}$ and $\rho$ according to a perturbation distribution $\Gamma$. Under assumption (A2) on the functions $\tilde{f}_1, \ldots, \tilde{f}_n$ and their expectations $f_1, \ldots, f_n$, we have, for all $x \in \mathbb{R}^p$,*

$$\mathbb{E}_{i,\rho}[\|\nabla \tilde{f}_i(x, \rho) - \nabla f_i(x^*)\|^2] \le 4L(f(x) - f(x^*)) + 2\sigma_p^2.$$

*Proof.* We have

$$\|\nabla \tilde{f}_i(x, \rho) - \nabla f_i(x^*)\|^2$$
$$\le 2\|\nabla \tilde{f}_i(x, \rho) - \nabla \tilde{f}_i(x^*, \rho)\|^2 + 2\|\nabla \tilde{f}_i(x^*, \rho) - \nabla f_i(x^*)\|^2$$
$$\le 4L(\tilde{f}_i(x, \rho) - \tilde{f}_i(x^*, \rho) - \langle \nabla \tilde{f}_i(x^*, \rho), x - x^* \rangle) + 2\|\nabla \tilde{f}_i(x^*, \rho) - \nabla f_i(x^*)\|^2.$$

The first inequality comes from the simple relation $\|u + v\|^2 + \|u - v\|^2 = 2\|u\|^2 + 2\|v\|^2$. The second inequality follows from the smoothness of $\tilde{f}_i(\cdot, \rho)$, in particular we used the classical relation

$$g(y) \ge g(x) + \langle \nabla g(x), y - x \rangle + \frac{1}{2L}\|\nabla g(y) - \nabla g(x)\|^2,$$

which is known to hold for any convex and $L$-smooth function $g$ (see, *e.g.*, [25, Theorem 2.1.5]). The result follows by taking expectations on $i$ and $\rho$ and noting that $\mathbb{E}_{i,\rho}[\nabla \tilde{f}_i(x^*, \rho)] = \nabla f(x^*) = 0$, as well as the definition of $\sigma_p^2$. $\qquad\square$

We now proceed with the proof of Proposition 1.

*Proof.* Define the quantities

$$A_t = \frac{1}{n}\sum_{i=1}^{n}\|z_i^t - z_i^*\|^2$$

$$\text{and} \quad B_t = \frac{1}{2}\|x_t - x^*\|^2.$$

The proof successively describes recursions on $A_t$, $B_t$, and eventually $C_t$.

**Recursion on $A_t$.** We have

$$A_t - A_{t-1} = \frac{1}{n}(\|z_{i_t}^t - z_{i_t}^*\|^2 - \|z_{i_t}^{t-1} - z_{i_t}^*\|^2)$$

$$= \frac{1}{n}\left(\left\|(1-\alpha_t)(z_{i_t}^{t-1} - z_{i_t}^*) + \alpha_t\left(x_{t-1} - \frac{1}{\mu}\nabla \tilde{f}_{i_t}(x_{t-1}, \rho_t) - z_{i_t}^*\right)\right\|^2 - \|z_{i_t}^{t-1} - z_{i_t}^*\|^2\right)$$

$$= \frac{1}{n}\left(-\alpha_t\|z_{i_t}^{t-1} - z_{i_t}^*\|^2 + \alpha_t\left\|x_{t-1} - \frac{1}{\mu}\nabla \tilde{f}_{i_t}(x_{t-1}, \rho_t) - z_{i_t}^*\right\|^2 - \alpha_t(1-\alpha_t)\|v_t\|^2\right), \tag{22}$$

where we first use the definition of $z_i^t$ in (3), then the relation $\|(1-\lambda)u + \lambda v\|^2 = (1-\lambda)\|u\|^2 + \lambda\|v\|^2 - \lambda(1-\lambda)\|u-v\|^2$, and the definition of $v_t$ given in (7). A similar relation is derived in the proof of SDCA without duality [28]. Using the definition of $z_i^*$, the second term can be expanded as

$$\left\|x_{t-1} - \frac{1}{\mu}\nabla \tilde{f}_{i_t}(x_{t-1}, \rho_t) - z_{i_t}^*\right\|^2 = \left\|x_{t-1} - x^* - \frac{1}{\mu}(\nabla \tilde{f}_{i_t}(x_{t-1}, \rho_t) - \nabla f_{i_t}(x^*))\right\|^2$$

$$= \|x_{t-1} - x^*\|^2 - \frac{2}{\mu}\langle x_{t-1} - x^*, \nabla \tilde{f}_{i_t}(x_{t-1}, \rho_t) - \nabla f_{i_t}(x^*)\rangle$$

$$+ \frac{1}{\mu^2}\left\|\nabla \tilde{f}_{i_t}(x_{t-1}, \rho_t) - \nabla f_{i_t}(x^*)\right\|^2. \tag{23}$$

We then take conditional expectations with respect to $\mathcal{F}_{t-1}$, defined in Section 2. Unless otherwise specified, we will simply write $\mathbb{E}[\cdot]$ instead of $\mathbb{E}[\cdot|\mathcal{F}_{t-1}]$ for these conditional expectations in the rest of the proof.

$$\mathbb{E}\left[\left\|x_{t-1}-\frac{1}{\mu}\nabla\tilde{f}_{i_t}(x_{t-1},\rho_t)-z_{i_t}^*\right\|^2\right] \leq \|x_{t-1}-x^*\|^2 - \frac{2}{\mu}\langle x_{t-1}-x^*, \nabla f(x_{t-1})\rangle$$

$$+\frac{4L}{\mu^2}(f(x_{t-1})-f(x^*)) + \frac{2\sigma_p^2}{\mu^2}$$

$$\leq \|x_{t-1}-x^*\|^2 - \frac{2}{\mu}(f(x_{t-1})-f(x^*) + \frac{\mu}{2}\|x_{t-1}-x^*\|^2)$$

$$+\frac{4L}{\mu^2}(f(x_{t-1})-f(x^*)) + \frac{2\sigma_p^2}{\mu^2}$$

$$=\frac{2(2\kappa-1)}{\mu}(f(x_{t-1})-f(x^*)) + \frac{2\sigma_p^2}{\mu^2},$$

where we used $\mathbb{E}[\nabla f_{i_t}(x^*)] = \nabla f(x^*) = 0$, Lemma B.1, and the $\mu$-strong convexity of $f$. Taking expectations on the previous relation on $A_t$ yields

$$\mathbb{E}[A_t - A_{t-1}] = -\frac{\alpha_t}{n}A_{t-1} + \frac{\alpha_t}{n}\mathbb{E}\left[\left\|x_{t-1}-\frac{1}{\mu}\nabla\tilde{f}_{i_t}(x_{t-1},\rho_t)-z_{i_t}^*\right\|^2\right] - \frac{\alpha_t(1-\alpha_t)}{n}\mathbb{E}[\|v_t\|^2]$$

$$\leq -\frac{\alpha_t}{n}A_{t-1} + \frac{2\alpha_t(2\kappa-1)}{n\mu}(f(x_{t-1})-f(x^*)) - \frac{\alpha_t(1-\alpha_t)}{n}\mathbb{E}[\|v_t\|^2] + \frac{2\alpha_t\sigma_p^2}{n\mu^2}.$$

$$(24)$$

**Recursion on $B_t$.** Separately, we have

$$\|x_t - x^*\|^2 = \left\|x_{t-1} - x^* + \frac{\alpha_t}{n}v_t\right\|^2$$

$$= \|x_{t-1}-x^*\|^2 + \frac{2\alpha_t}{n}\langle x_{t-1}-x^*, v_t\rangle + \left(\frac{\alpha_t}{n}\right)^2\|v_t\|^2$$

$$\mathbb{E}[\|x_t - x^*\|^2] = \|x_{t-1}-x^*\|^2 - \frac{2\alpha_t}{n\mu}\langle x_{t-1}-x^*, \nabla f(x_{t-1})\rangle + \left(\frac{\alpha_t}{n}\right)^2\mathbb{E}[\|v_t\|^2]$$

$$\leq \|x_{t-1}-x^*\|^2 - \frac{2\alpha_t}{n\mu}(f(x_{t-1})-f(x^*) + \frac{\mu}{2}\|x_{t-1}-x^*\|^2) + \left(\frac{\alpha_t}{n}\right)^2\mathbb{E}[\|v_t\|^2],$$

using that $\mathbb{E}[v_t] = -\frac{1}{\mu}\nabla f(x_{t-1})$ and the strong convexity of $f$. This gives

$$\mathbb{E}[B_t - B_{t-1}] \leq -\frac{\alpha_t}{n}B_{t-1} - \frac{\alpha_t}{n\mu}(f(x_{t-1})-f(x^*)) + \frac{1}{2}\left(\frac{\alpha_t}{n}\right)^2\mathbb{E}[\|v_t\|^2]. \qquad (25)$$

**Recursion on $C_t$.** If we consider $C_t = p_t A_t + B_t$ and $C'_{t-1} = p_t A_{t-1} + B_{t-1}$, combining (24) and (25) yields

$$\mathbb{E}[C_t - C'_{t-1}] \leq$$

$$-\frac{\alpha_t}{n}C'_{t-1} + \frac{2\alpha_t}{n\mu}(p_t(2\kappa-1)-\frac{1}{2})(f(x_{t-1})-f(x^*)) + \frac{\alpha_t}{n}\left(\frac{\alpha_t}{2n} - p_t(1-\alpha_t)\right)\mathbb{E}[\|v_t\|^2] + \frac{2\alpha_t p_t\sigma_p^2}{n\mu^2}.$$

If we take $p_t = \frac{\alpha_t}{n}$, and if $(\alpha_t)_{t\geq 1}$ is a decreasing sequence satisfying (9), then the factors in front of $f(x_{t-1}) - f(x^*)$ and $\mathbb{E}[\|v_t\|^2]$ are non-positive and we get

$$\mathbb{E}[C_t] \leq \left(1 - \frac{\alpha_t}{n}\right)C'_{t-1} + 2\left(\frac{\alpha_t}{n}\right)^2\frac{\sigma_p^2}{\mu^2}.$$

Finally, since $\alpha_t \leq \alpha_{t-1}$, we have $C'_{t-1} \leq C_{t-1}$. After taking total expectations on $\mathcal{F}_{t-1}$, we are left with the desired recursion.

$\square$

## B.2 Proof of Theorem 2 (Convergence of $C_t$ under decreasing step-sizes)

We prove the theorem with more general step-sizes:

**Theorem B.1** (Convergence of Lyapunov function)**.** *Let the sequence of step-sizes $(\alpha_t)_{t\geq 1}$ be defined by $\alpha_t = \frac{\beta n}{\gamma + t}$ with $\beta > 1$ and $\gamma \geq 0$ such that $\alpha_1$ satisfies (9). For all $t \geq 0$, it holds that*

$$\mathbb{E}[C_t] \leq \frac{\nu}{\gamma + t + 1} \quad where \quad \nu := \max\left\{ \frac{2\beta^2 \sigma_p^2}{\mu^2(\beta - 1)}, (\gamma + 1)C_0 \right\}. \qquad (26)$$

In particular, taking $\beta = 2$ as in Theorem 2 can only improve the constant $\nu$ in the convergence rate.

*Proof.* Let us proceed by induction. We have $C_0 \leq \nu/(\gamma + 1)$ by definition of $\nu$. For $t \geq 1$,

$$\mathbb{E}[C_t] \leq \left(1 - \frac{\alpha_t}{n}\right) \mathbb{E}[C_{t-1}] + 2\left(\frac{\alpha_t}{n}\right)^2 \frac{\sigma_p^2}{\mu^2}$$

$$\leq \left(1 - \frac{\beta}{\hat{t}}\right) \frac{\nu}{\hat{t}} + \frac{2\beta^2 \sigma_p^2}{\hat{t}^2 \mu^2} \quad \text{(with } \hat{t} := \gamma + t)$$

$$= \left(\frac{\hat{t} - \beta}{\hat{t}^2}\right) \nu + \frac{2\beta^2 \sigma_p^2}{\hat{t}^2 \mu^2}$$

$$= \left(\frac{\hat{t} - 1}{\hat{t}^2}\right) \nu - \left(\frac{\beta - 1}{\hat{t}^2}\right) \nu + \frac{2\beta^2 \sigma_p^2}{\hat{t}^2 \mu^2}$$

$$\leq \left(\frac{\hat{t} - 1}{\hat{t}^2}\right) \nu \leq \frac{\nu}{\hat{t} + 1},$$

where the last two inequalities follow from the definition of $\nu$ and from $\hat{t}^2 \geq (\hat{t} + 1)(\hat{t} - 1)$. $\quad\square$

## B.3 Proof of Theorem 3 (Convergence in function values under iterate averaging)

*Proof.* From the proof of Proposition 1, we have

$$\mathbb{E}[C_t] \leq \left(1 - \frac{\alpha_t}{n}\right) \mathbb{E}[C_{t-1}] + \frac{2\alpha_t}{n\mu}\left(\frac{\alpha_t}{n}(2\kappa - 1) - \frac{1}{2}\right) \mathbb{E}[f(x_{t-1}) - f(x^*)] + 2\left(\frac{\alpha_t}{n}\right)^2 \frac{\sigma_p^2}{\mu^2}.$$

The result holds because the choice of step-sizes $(\alpha_t)_{t\geq 1}$ safisfies the assumptions of Proposition 1. With our new choice of step-sizes, we have the stronger bound

$$\frac{\alpha_t}{n}(2\kappa - 1) - \frac{1}{2} \leq -\frac{1}{4}.$$

After rearranging, we obtain

$$\frac{\alpha_t}{2n\mu} \mathbb{E}[f(x_{t-1}) - f(x^*)] \leq \left(1 - \frac{\alpha_t}{n}\right) \mathbb{E}[C_{t-1}] - \mathbb{E}[C_t] + 2\left(\frac{\alpha_t}{n}\right)^2 \frac{\sigma_p^2}{\mu^2}. \qquad (27)$$

Dividing by $\frac{\alpha_t}{2n\mu}$ gives

$$\mathbb{E}[f(x_{t-1}) - f(x^*)] \leq 2\mu\left[\left(\frac{n}{\alpha_t} - 1\right) \mathbb{E}[C_{t-1}] - \frac{n}{\alpha_t} \mathbb{E}[C_t]\right] + 4\frac{\alpha_t}{n}\frac{\sigma_p^2}{\mu}$$

$$= \mu\left((\gamma + t - 2)\mathbb{E}[C_{t-1}] - (\gamma + t)\mathbb{E}[C_t]\right) + \frac{8}{\gamma + t}\frac{\sigma_p^2}{\mu}.$$

Multiplying by $(\gamma + t - 1)$ yields

$$(\gamma + t - 1)\mathbb{E}[f(x_{t-1}) - f(x^*)]$$

$$\leq \mu\left((\gamma + t - 1)(\gamma + t - 2)\mathbb{E}[C_{t-1}] - (\gamma + t)(\gamma + t - 1)\mathbb{E}[C_t]\right) + \frac{8(\gamma + t - 1)}{\gamma + t}\frac{\sigma_p^2}{\mu}$$

$$\leq \mu\left((\gamma + t - 1)(\gamma + t - 2)\mathbb{E}[C_{t-1}] - (\gamma + t)(\gamma + t - 1)\mathbb{E}[C_t]\right) + \frac{8\sigma_p^2}{\mu}.$$

By summing the above inequality from $t = 1$ to $t = T$, we have a telescoping sum that simplifies as follows:

$$\mathbb{E}\left[\sum_{t=1}^{T}(\gamma+t-1)(f(x_{t-1})-f(x^*))\right] \leq \mu\left(\gamma(\gamma-1)C_0 - (\gamma+T)(\gamma+T-1)\,\mathbb{E}[C_T]\right) + \frac{8T\sigma_p^2}{\mu}$$

$$\leq \mu\gamma(\gamma-1)C_0 + \frac{8T\sigma_p^2}{\mu}.$$

Dividing by $\sum_{t=1}^{T}(\gamma+t-1) = (2T\gamma + T(T-1))/2$ and using Jensen's inequality on $f(\bar{x}_T)$ gives the desired result. $\qquad\square$

## C  Proofs for Composite Objectives and Non-Uniform Sampling (Appendix A)

We recall here the updates to the lower bounds $d_i^t$ in the setting of this section, which are analogous to (6) but with non-uniform weights and stochastic perturbations,: for $i = i_t$, we have

$$d_i^t(x) = \left(1 - \frac{\alpha_t}{q_i n}\right)d_i^{t-1}(x) + \frac{\alpha_t}{q_i n}\left(\tilde{f}_i(x_{t-1}, \rho_t) + \langle\nabla\tilde{f}_i(x_{t-1}, \rho_t), x - x_{t-1}\rangle + \frac{\mu}{2}\|x - x_{t-1}\|^2\right), \quad (28)$$

and $d_i^t(x) = d_i^{t-1}(x)$ otherwise.

### C.1  Proof of Lemma 4 (Bound on the iterates)

*Proof.* Let $F_t(x) := \frac{1}{n}\sum_{i=1}^{n}f_i^t(x) + h(x)$, where $f_i^0(x) = \tilde{f}_i(x, \tilde{\rho}_i)$ (where $\tilde{\rho}_i$ is used in (16)), and $f_i^t$ is updated analogously to $d_i^t$ as follows:

$$f_i^t(x) = \begin{cases} (1 - \frac{\alpha_t}{q_i n})f_i^{t-1}(x) + \frac{\alpha_t}{q_i n}\tilde{f}_i(x, \rho_t), & \text{if } i = i_t \\ f_i^{t-1}(x), & \text{otherwise.} \end{cases}$$

By induction, we have

$$F_t(x^*) \geq D_t(x^*) \geq D_t(x_t) + \frac{\mu}{2}\|x_t - x^*\|^2, \quad (29)$$

where the last inequality follows from the $\mu$-strong convexity of $D_t$ and the fact that $x_t$ is its minimizer.

Again by induction, we now show that $\mathbb{E}[F_t(x^*)] = F(x^*)$. Indeed, we have $\mathbb{E}[F_0(x^*)] = F(x^*)$ by construction, then

$$F_t(x^*) = F_{t-1}(x^*) + \frac{\alpha_t}{q_i n^2}(\tilde{f}_{i_t}(x^*, \rho_t) - f_{i_t}^{t-1}(x^*))$$

$$\mathbb{E}[F_t(x^*)|\mathcal{F}_{t-1}] = F_{t-1}(x^*) + \frac{\alpha_t}{n}\left(f(x^*) - \frac{1}{n}\sum_{i=1}^{n}f_i^{t-1}(x^*)\right)$$

$$= F_{t-1}(x^*) + \frac{\alpha_t}{n}(F(x^*) - F_{t-1}(x^*)),$$

After taking total expectations and using the induction hypothesis, we obtain $\mathbb{E}[F_t(x^*)] = F(x^*)$, and the result follows from (29). $\qquad\square$

### C.2  Proof of Proposition 5 (Recursion on Lyapunov function $C_t^q$)

We begin by presenting a lemma that plays a similar role to Lemma B.1 in our proof, but considers the composite objective and takes into account the new strong convexity and non-uniformity assumptions.

**Lemma C.1.** *Let $i \sim q$, where $q$ is the sampling distribution, and $\rho$ be a random perturbation. Under assumptions (A4-5) on the functions $\tilde{f}_1, \ldots, \tilde{f}_n$ and their expectations $f_1, \ldots, f_n$, we have, for all $x \in \mathbb{R}^p$,*

$$\mathbb{E}_{i,\rho}\left[\frac{1}{(q_i n)^2}\|\nabla\tilde{f}_i(x, \rho) - \mu x - (\nabla f_i(x^*) - \mu x^*)\|^2\right] \leq 4L_q(F(x) - F(x^*)) + 2\sigma_q^2,$$

*with $L_q = \max_i \frac{L_i - \mu}{q_i n}$ and $\sigma_q^2 = \frac{1}{n}\sum_i \frac{\sigma_i^2}{q_i n}$.*

*Proof.* Since $\tilde{f}_i(\cdot, \rho)$ is $\mu$-strongly convex and $L_i$-smooth, we have that $\tilde{f}_i(\cdot, \rho) - \frac{\mu}{2}\|\cdot\|^2$ is convex and $(L_i - \mu)$-smooth [this is a straightforward consequence of 25, Eq. 2.1.9 and 2.1.22]. Then, by denoting by $F_i$ the quantity $2\|\nabla\tilde{f}_i(x^*, \rho) - \nabla f_i(x^*)\|^2$, we have

$$\|\nabla\tilde{f}_i(x, \rho) - \mu x - (\nabla f_i(x^*) - \mu x^*)\|^2$$

$$\leq 2\|\nabla\tilde{f}_i(x, \rho) - \mu x - (\nabla\tilde{f}_i(x^*, \rho) - \mu x^*)\|^2 + 2\|\nabla\tilde{f}_i(x^*, \rho) - \nabla f_i(x^*)\|^2$$

$$\leq 4(L_i - \mu)\left(\tilde{f}_i(x, \rho) - \frac{\mu}{2}\|x\|^2 - \tilde{f}_i(x^*, \rho) + \frac{\mu}{2}\|x^*\|^2 - \langle\nabla\tilde{f}_i(x^*, \rho) - \mu x^*, x - x^*\rangle\right) + F_i$$

$$= 4(L_i - \mu)\left(\tilde{f}_i(x, \rho) - \tilde{f}_i(x^*, \rho) - \langle\nabla\tilde{f}_i(x^*, \rho), x - x^*\rangle - \frac{\mu}{2}\|x - x^*\|^2\right) + F_i$$

$$\leq 4(L_i - \mu)\left(\tilde{f}_i(x, \rho) - \tilde{f}_i(x^*, \rho) - \langle\nabla\tilde{f}_i(x^*, \rho), x - x^*\rangle\right) + F_i.$$

The first inequality comes from the classical relation $\|u + v\|^2 + \|u - v\|^2 = 2\|u\|^2 + 2\|v\|^2$. The second inequality follows from the convexity and $(L_i - \mu)$-smoothness of $\tilde{f}_i(\cdot, \rho) - \frac{\mu}{2}\|\cdot\|^2$. Dividing by $(q_i n)^2$ and taking expectations yields

$$\mathbb{E}_{i,\rho}\left[\frac{1}{(q_i n)^2}\|\nabla\tilde{f}_i(x, \rho) - \mu x - (\nabla f_i(x^*) - \mu x^*)\|^2\right]$$

$$\leq 4\sum_{i=1}^{n}\frac{q_i(L_i - \mu)}{(q_i n)^2}(f_i(x) - f_i(x^*) - \langle\nabla f_i(x^*), x - x^*\rangle) + 2\sum_{i=1}^{n}\frac{q_i}{(q_i n)^2}\sigma_i^2$$

$$= 4\frac{1}{n}\sum_{i=1}^{n}\frac{L_i - \mu}{q_i n}(f_i(x) - f_i(x^*) - \langle\nabla f_i(x^*), x - x^*\rangle) + 2\frac{1}{n}\sum_{i=1}^{n}\frac{\sigma_i^2}{q_i n}$$

$$\leq 4L_q(f(x) - f(x^*) - \langle\nabla f(x^*), x - x^*\rangle) + 2\sigma_q^2$$

$$\leq 4L_q(f(x) - f(x^*) + h(x) - h(x^*)) + 2\sigma_q^2 = 4L_q(F(x) - F(x^*)) + 2\sigma_q^2,$$

where the last inequality follows from the optimality of $x^*$, which implies that $-\nabla f(x^*) \in \partial h(x^*)$, and in turn implies $\langle-\nabla f(x^*), x - x^*\rangle \leq h(x) - h(x^*)$ by convexity of $h$. $\qquad\square$

We can now proceed with the proof of Proposition 5.

*Proof.* Define the quantities

$$A_t = \frac{1}{n}\sum_{i=1}^{n}\frac{1}{q_i n}\|z_i^t - z_i^*\|^2$$

$$\text{and} \quad B_t = F(x^*) - D_t(x_t).$$

The proof successively describes recursions on $A_t$, $B_t$, and eventually $C_t$ (we drop the superscript in $C_t^q$ for simplicity), using the same approach as for the proof of Proposition 1.

**Recursion on $A_t$.** Using similar techniques as in the proof of Proposition 1, we have

$$A_t - A_{t-1}$$

$$= \frac{1}{n}\left(\frac{1}{q_{i_t}n}\|z_{i_t}^t - z_{i_t}^*\|^2 - \frac{1}{q_{i_t}n}\|z_{i_t}^{t-1} - z_{i_t}^*\|^2\right)$$

$$= \frac{1}{n}\left(\frac{1}{q_{i_t}n}\left\|\left(1 - \frac{\alpha_t}{q_{i_t}n}\right)(z_{i_t}^{t-1} - z_{i_t}^*) + \frac{\alpha_t}{q_{i_t}n}\left(x_{t-1} - \frac{1}{\mu}\nabla\tilde{f}_{i_t}(x_{t-1}, \rho_t) - z_{i_t}^*\right)\right\|^2 - \frac{1}{q_{i_t}n}\|z_{i_t}^{t-1} - z_{i_t}^*\|^2\right)$$

$$= \frac{1}{n}\left(-\frac{\alpha_t}{(q_{i_t}n)^2}\|z_{i_t}^{t-1} - z_{i_t}^*\|^2 + \frac{\alpha_t}{(q_{i_t}n)^2}\left\|x_{t-1} - \frac{1}{\mu}\nabla\tilde{f}_{i_t}(x_{t-1}, \rho_t) - z_{i_t}^*\right\|^2 - \frac{\alpha_t}{(q_{i_t}n)^2}\left(1 - \frac{\alpha_t}{q_{i_t}n}\right)\|v_{i_t}^t\|^2\right),$$

where $v_i^t := x_{t-1} - \frac{1}{\mu}\nabla\tilde{f}_i(x_{t-1}, \rho_t) - z_i^{t-1}$. Taking conditional expectations w.r.t. $\mathcal{F}_{t-1}$ and using Lemma C.1 to bound the second term yields

$$\mathbb{E}[A_t - A_{t-1}] \leq$$

$$-\frac{\alpha_t}{n}A_{t-1} + \frac{4\alpha_t L_q}{n\mu^2}(F(x_{t-1}) - F(x^*)) + \frac{2\alpha_t\sigma_q^2}{n\mu^2} - \frac{1}{n}\sum_{i=1}^{n}\left(\frac{\alpha_t}{n}\frac{1}{q_i n}\left(1 - \frac{\alpha_t}{q_i n}\right)\|v_i^t\|^2\right) \quad (30)$$

**Recursion on $B_t$.** We start by using a lemma from the proof of MISO-Prox [18, Lemma D.4], which only relies on the form of $D_t$ and the fact that $x_t$ minimizes it, and thus holds in our setting:

$$D_t(x_t) \geq D_t(x_{t-1}) - \frac{\mu}{2}\|\bar{z}_t - \bar{z}_{t-1}\|^2$$

$$= D_t(x_{t-1}) - \frac{\mu}{2(q_{i_t}n)^2}\left(\frac{\alpha_t}{n}\right)^2 \|v_{i_t}^t\|^2 \tag{31}$$

We then expand $D_t(x_{t-1})$ using (28) as follows:

$$D_t(x_{t-1}) = D_{t-1}(x_{t-1}) + \frac{\alpha_t}{n}\frac{1}{q_{i_t}n}\left(\tilde{f}_{i_t}(x_{t-1},\rho_t) - d_{i_t}^{t-1}(x_{t-1})\right)$$

$$= D_{t-1}(x_{t-1}) + \frac{\alpha_t}{n}\frac{1}{q_{i_t}n}\left(\tilde{f}_{i_t}(x_{t-1},\rho_t) + h(x_{t-1}) - d_{i_t}^{t-1}(x_{t-1}) - h(x_{t-1})\right).$$

After taking conditional expections w.r.t. $\mathcal{F}_{t-1}$, (31) becomes

$$\mathbb{E}[D_t(x_t)] \geq D_{t-1}(x_{t-1}) + \frac{\alpha_t}{n}(F(x_{t-1}) - D_{t-1}(x_{t-1})) - \frac{\mu}{2n}\sum_{i=1}^n\left(\frac{\alpha_t}{n}\right)^2\frac{1}{q_in}\|v_i^t\|^2.$$

Subtracting $F(x^*)$ and rearranging yields

$$\mathbb{E}[B_t - B_{t-1}] \leq -\frac{\alpha_t}{n}B_{t-1} - \frac{\alpha_t}{n}(F(x_{t-1}) - F(x^*)) + \frac{\mu}{2n}\sum_{i=1}^n\left(\frac{\alpha_t}{n}\right)^2\frac{1}{q_in}\|v_i^t\|^2. \tag{32}$$

**Recursion on $C_t$.** If we consider $C_t = \mu p_t A_t + B_t$ and $C'_{t-1} = \mu p_t A_{t-1} + B_{t-1}$, combining (30) and (32) yields

$$\mathbb{E}[C_t - C'_{t-1}] \leq -\frac{\alpha_t}{n}C'_{t-1} + \frac{2\alpha_t}{n}\left(\frac{2p_t L_q}{\mu} - \frac{1}{2}\right)(F(x_{t-1}) - F(x^*)) + \frac{\mu\alpha_t}{n^2}\sum_{i=1}^n\frac{\delta_i^t}{q_in}\|v_i^t\|^2 + \frac{2\alpha_t p_t \sigma_q^2}{n\mu}, \tag{33}$$

with

$$\delta_i^t = \frac{\alpha_t}{2n} - p_t\left(1 - \frac{\alpha_t}{q_in}\right).$$

If we take $p_t = \frac{\alpha_t}{n}$, and if $(\alpha_t)_{t\geq 1}$ is a decreasing sequence satisfying (19), then we obtain the desired recursion after noticing that $C'_{t-1} \leq C_{t-1}$ and taking total expectations on $\mathcal{F}_{t-1}$. $\qquad\square$

Note that if we take

$$\alpha_1 \leq \min\left\{\frac{nq_{\min}}{2}, \frac{n\mu}{8L_q}\right\},$$

then (33) yields

$$\mathbb{E}\left[\frac{C_t^q}{\mu}\right] \leq \left(1 - \frac{\alpha_t}{n}\right)\mathbb{E}\left[\frac{C_{t-1}^q}{\mu}\right] - \frac{\alpha_t}{2n\mu}(F(x_{t-1}) - F(x^*)) + 2\left(\frac{\alpha_t}{n}\right)^2\frac{\sigma_q^2}{\mu^2}.$$

This relation takes the same form as Eq. (27), hence it is straightforward to adapt the proof of Theorem 3 to this setting, and the same iterate averaging scheme applies.

## D  Complexity Analysis of SGD

In this section, we provide a proof of a simple result for SGD in the smooth case, giving a recursion that depends on a variance condition at the optimum (in contrast to [5, 24] where this condition needs to hold everywhere), for a more natural comparison with S-MISO.

**Proposition D.1** (Simple SGD recursion with variance at optimum). *Under assumptions (A1) and (A2), if $\eta_t \leq 1/2L$, then the SGD recursion $x_t := x_{t-1} - \eta_t\nabla\tilde{f}_{i_t}(x_{t-1},\rho_t)$ satisfies*

$$B_t \leq (1 - \mu\eta_t)B_{t-1} + \eta_t^2\sigma_{tot}^2,$$

*where $B_t := \frac{1}{2}\mathbb{E}[\|x_t - x^*\|^2]$ and $\sigma_{tot}$ is such that*

$$\mathbb{E}_{i,\rho}\left[\|\nabla\tilde{f}_i(x^*,\rho)\|^2\right] \leq \sigma_{tot}^2.$$

*Proof.* We have

$$\|x_t - x^*\|^2 = \|x_{t-1} - x^*\|^2 - 2\eta_t \langle \nabla \tilde{f}_{i_t}(x_{t-1}, \rho_t), x_{t-1} - x^* \rangle + \eta_t^2 \|\nabla \tilde{f}_{i_t}(x_{t-1}, \rho_t)\|^2$$

$$\leq \|x_{t-1} - x^*\|^2 - 2\eta_t \langle \nabla \tilde{f}_{i_t}(x_{t-1}, \rho_t), x_{t-1} - x^* \rangle$$

$$+ 2\eta_t^2 \|\nabla \tilde{f}_{i_t}(x_{t-1}, \rho_t) - \nabla \tilde{f}_{i_t}(x^*, \rho_t)\|^2 + 2\eta_t^2 \|\nabla \tilde{f}_{i_t}(x^*, \rho_t)\|^2$$

$$\mathbb{E}\left[\|x_t - x^*\|^2\right] \leq \|x_{t-1} - x^*\|^2 - 2\eta_t \langle \nabla f(x_{t-1}), x_{t-1} - x^* \rangle$$

$$+ 2\eta_t^2 \, \mathbb{E}_{i_t, \rho_t}\left[\|\nabla \tilde{f}_{i_t}(x_{t-1}, \rho_t) - \nabla \tilde{f}_{i_t}(x^*, \rho_t)\|^2\right] + 2\eta_t^2 \, \mathbb{E}_{i_t, \rho_t}\left[\|\nabla \tilde{f}_{i_t}(x^*, \rho_t)\|^2\right]$$

$$(*) \leq \|x_{t-1} - x^*\|^2 - 2\eta_t \left(f(x_{t-1}) - f(x^*) + \frac{\mu}{2}\|x_{t-1} - x^*\|^2\right)$$

$$+ 4L\eta_t^2 \left(f(x_{t-1}) - f(x^*)\right) + 2\eta_t^2 \sigma_{\text{tot}}^2$$

$$= (1 - \mu\eta_t)\|x_{t-1} - x^*\|^2 - 2\eta_t(1 - 2L\eta_t)(f(x_{t-1}) - f(x^*)) + 2\eta_t^2 \sigma_{\text{tot}}^2,$$

where the expectation is taken with respect to the filtration $\mathcal{F}_{t-1}$ and the inequality $(*)$ follows from the strong convexity of $f$ and $\mathbb{E}_{i,\rho}[\|\nabla \tilde{f}_{i_t}(x_{t-1}, \rho_t) - \nabla \tilde{f}_{i_t}(x^*, \rho_t)\|^2]$ is bounded by $2L(f(x_{t-1}) - f(x^*))$ as in the proof of Lemma B.1. When $\eta_t \leq 1/2L$, the second term is non-positive and we obtain the desired result after taking total expectations. $\square$

Note that when $\eta_t \leq 1/4L$, we have

$$\mathbb{E}\left[\|x_t - x^*\|^2\right] \leq (1 - \mu\eta_t)\mathbb{E}\left[\|x_{t-1} - x^*\|^2\right] - \eta_t(f(x_{t-1}) - f(x^*)) + 2\eta_t^2 \sigma_{\text{tot}}^2.$$

This takes a similar form to Eq. (27), and one can use the same iterate averaging scheme as Theorem 3 with step-sizes $\eta_t = 2/\mu(\gamma + t)$ by adapting the proof.

We now give a similar recursion for the proximal SGD algorithm (see, *e.g.*, [11]). This allows us to apply the results of Theorem 2 and the step-size strategy mentioned in Section 3.

**Proposition D.2** (Simple recursion for proximal SGD with variance at optimum). *Under assumptions (A1) and (A2), if $\eta_t \leq 1/2L$, then the proximal SGD recursion*

$$x_t := \text{prox}_{\eta_t h}(x_{t-1} - \eta_t \nabla \tilde{f}_{i_t}(x_{t-1}, \rho_t))$$

*satisfies*

$$B_t \leq (1 - \mu\eta_t)B_{t-1} + \eta_t^2 \sigma_{tot}^2,$$

*where $B_t := \frac{1}{2}\mathbb{E}[\|x_t - x^*\|^2]$ and $\sigma_{tot}$ is such that*

$$\mathbb{E}_{i,\rho}\left[\|\nabla \tilde{f}_i(x^*, \rho) - \nabla f(x^*)\|^2\right] \leq \sigma_{tot}^2.$$

*Proof.* We have

$$\|x_t - x^*\|^2$$

$$= \|\text{prox}_{\eta_t h}(x_{t-1} - \eta_t \nabla \tilde{f}_{i_t}(x_{t-1}, \rho_t)) - \text{prox}_{\eta_t h}(x^* - \eta_t \nabla f(x^*))\|^2$$

$$\leq \|x_{t-1} - \eta_t \nabla \tilde{f}_{i_t}(x_{t-1}, \rho_t) - x^* + \eta_t \nabla f(x^*)\|^2$$

$$= \|x_{t-1} - x^*\|^2 - 2\eta_t \langle \nabla \tilde{f}_{i_t}(x_{t-1}, \rho_t) - \nabla f(x^*), x_{t-1} - x^* \rangle + \eta_t^2 \|\nabla \tilde{f}_{i_t}(x_{t-1}, \rho_t) - \nabla f(x^*)\|^2$$

$$\leq \|x_{t-1} - x^*\|^2 - 2\eta_t \langle \nabla \tilde{f}_{i_t}(x_{t-1}, \rho_t) - \nabla f(x^*), x_{t-1} - x^* \rangle$$

$$+ 2\eta_t^2 \|\nabla \tilde{f}_{i_t}(x_{t-1}, \rho_t) - \nabla \tilde{f}_{i_t}(x^*, \rho_t)\|^2 + 2\eta_t^2 \|\nabla \tilde{f}_{i_t}(x^*, \rho_t) - \nabla f(x^*)\|^2,$$

where the first equality follows from the optimality of $x^*$ and the following inequality follows from the non-expansiveness of proximal operators. Taking conditional expectations on $\mathcal{F}_{t-1}$ yields

$$\mathbb{E}\left[\|x_t - x^*\|^2 | \mathcal{F}_{t-1}\right]$$

$$\leq \|x_{t-1} - x^*\|^2 - 2\eta_t \langle \nabla f(x_{t-1}) - \nabla f(x^*), x_{t-1} - x^* \rangle$$

$$+ 2\eta_t^2 \, \mathbb{E}_{i_t, \rho_t}\left[\|\nabla \tilde{f}_{i_t}(x_{t-1}, \rho_t) - \nabla \tilde{f}_{i_t}(x^*, \rho_t)\|^2\right] + 2\eta_t^2 \, \mathbb{E}_{i_t, \rho_t}\left[\|\nabla \tilde{f}_{i_t}(x^*, \rho_t) - \nabla f(x^*)\|^2\right]$$

$$(*) \leq \|x_{t-1} - x^*\|^2 - 2\eta_t \left(f(x_{t-1}) - f(x^*) + \frac{\mu}{2}\|x_{t-1} - x^*\|^2 - \langle \nabla f(x^*), x_{t-1} - x^* \rangle\right)$$

$$+ 4L\eta_t^2 \left(f(x_{t-1}) - f(x^*) - \langle \nabla f(x^*), x_{t-1} - x^* \rangle\right) + 2\eta_t^2 \sigma_{\text{tot}}^2$$

$$= (1 - \mu\eta_t)\|x_{t-1} - x^*\|^2 - 2\eta_t(1 - 2L\eta_t)(f(x_{t-1}) - f(x^*) - \langle \nabla f(x^*), x_{t-1} - x^* \rangle) + 2\eta_t^2 \sigma_{\text{tot}}^2,$$

Figure 4: Comparison of S-MISO and SGD with averaging, for different condition numbers (controlled by $\mu$) and different Dropout rates $\delta$. We begin step-size decay and averaging at epoch 3 (top) and 30 (bottom).

where inequality $(*)$ follows from the $\mu$-strong convexity of $f$ and $\mathbb{E}_{i,\rho}[\|\nabla \tilde{f}_{i_t}(x_{t-1}, \rho_t) - \nabla \tilde{f}_{i_t}(x^*, \rho_t)\|^2]$ is bounded by $2L(f(x_{t-1}) - f(x^*) - \langle \nabla f(x^*), x_{t-1} - x^* \rangle)$ as in the proof of Lemma C.1. By convexity of $f$, we have $f(x_{t-1}) - f(x^*) - \langle \nabla f(x^*), x_{t-1} - x^* \rangle \geq 0$, hence the second term is non-positive when $\eta_t \leq 1/2L$. We conclude by taking total expectations. $\square$

We note that Propositions D.1 and D.2 can be easily adapted to non-uniform sampling with sampling distribution $q$ and step-sizes $\eta_t/q_{i_t}n$, leading to step-size conditions $\eta_t \leq 1/2L_q$, with $L_q = \max_i \frac{L_i}{q_i n}$ and variance $\sigma_{q,tot}^2 = \mathbb{E}_{i,\rho}[\frac{1}{(q_i n)^2}\|\nabla \tilde{f}_i(x^*, \rho) - \nabla f(x^*)\|^2]$.

## E   Experiments with Averaging Scheme

Figure 4 shows a comparison of S-MISO and SGD with the averaging scheme proposed in Theorem 3 (see Appendix D for comments on how it applies to SGD), on the breast cancer dataset presented in Section 4, for different values of the regularization $\mu$ (and thus of the condition number $\kappa = L/\mu$), and Dropout rates $\delta$. We can see that the averaging scheme gives some small improvements for both methods, and that the improvements are more significant when the problem is more ill-conditioned (Figure 4, bottom). We note that the time at which we start averaging can have significant impact on the convergence, in particular, starting too early can significantly slow down the initial convergence, as commonly noticed for stochastic gradient methods (see, *e.g.*, [24]).