[Reviews · NeurIPS 2017]

Reviewer 1



The paper tackles the problem of variance reduction in the setting in which each example may have been randomly perturbed (e.g. for images, by small rotations, scalings, addition of noise, etc). The proposed algorithm is is based on the MISO/Finito algorithms, with the surrogate function associated with each (perturbed) example being a moving average of quadratic lower bounds, although, due to the permutations, these lower bounds are only approximate (the lower bound the particular current instance of the permutation on the current example, instead of the expectation over all permutations). They prove that the convergence rate of their algorithm, in contrast with SGD, depends only on the portion of the variance due to the permutations (\sigma_{\rho}) at the optimum, rather than the overall variance (which also includes a portion due to sampling of the examples). Finally, they close with a convincing set of experiments. While the algorithm itself seems a bit incremental, the setting is an important one, the algorithm is intuitive, and the results are complete, in the sense that both a good theoretical and experimental validation are included. One complaint I have involves the plots: the font sizes on the legends are very small, the yellow curve is too bright (not enough contrast with the white background), and the curves are differentiated by color alone, instead of by symbols or dashing, which makes them hard (or impossible) to read in black and white.

Reviewer 2



The paper proposes a method for optimization problems often found in machine learning tasks. The general loss function to minimize is of the form of a sum of smooth-convex functions associated with a convex regularization potential. The method is designed for the case of perturbation introduced in the data. Since the data sampling introduces a stochastic component Stochastic Gradient Descent (SGD) need of modifications for reducing the gradient variance [14,28]. In the case of perturbed data, such variance is magnified. The authors claim that the reported modifications to SGD for variance reductions limit their applicability for inline training because they requires of storing the full gradients or the dual variables. The reported SGD variant (S-MIMO) overcomes the mentioned limitations and preserves the accelerated characteristics of the SCG variants for batch processing. My main comment of the paper is that, in recent years, is has been reported many variants of SGD and it is hard to evaluate if the presented comparison is fair. Let me explain a possible experiment that is not included (among others). The method SVRG [14] requires of computing the full gradient after some m Stochastic Iterations, that assumes to have available all the data. That is the principal argument of the authors to say that it SVSG not be used for an inline training. However, is we conside the case of finite data., then each perturbed data can be seen as a new data. It could be interesting try with SVRS to train a problem using a epochs based strategy (each epoch uses different perturbations), and to compare its performance with S-MIMO. The same argument could be justify to use SVRG is a batch processing where the (full) gradient is updated after each batch. Now, suppose that the sketched idea for using SVRG in the optimization problems of the form (2) works; it justify a new paper proposing the “new method”? Do not in my opinion, do not in NIPS. Well this is the kind “variants” that saturate the literature. As resume, the authors should make an effort to compare their algorithm with more variants as possible, even if trivial modifications are required (as the sketched). Remark. I did not check the algebra of all the proof in the supplementary material, I could follow the text and seems correct.

Reviewer 3



The paper presents a variance reduction stochastic optimization algorithm for the case of perturbations of data that explicitly takes into account the fact that the variance among datapoints will be far higher than among the perturbations of a single one. Improvement is shown both theoretically and using simulations. I really like the motivation to handle the perturbation variance separately leading to improvements in convergence. The convergence results are all interesting (I was not able to fully verify their correctness). There seem to be (non-critical) typos in the derivation: refer line 428: is it (1-\alpha_t) instead of \alpha_t? I have read the rebuttal and would like to stay with the score.

Reviewer 4



Summary of the paper ==================== The paper considers the setting of finite-sum problems where each individual function undergoes a random pertuerbation. The authors devise an adapation of the finite-sums algorithm MISO\Finito, called S-MISO, for this setting, and provide a convergence analysis based on the noise (and the analytic) parameters of the model, along with supporting experiments on various datasets from different domains. Evaluation ========== The algorithm devised by the authors seems to effectively exploit the extra structure offered by this setting (in comparison to the generic stochastic optimization setting) - both theoretically where S-MISO is proven to improve upon SGD in a factor dependent on the overall variance and the average individual variance, and empirically, where S-MISO is shown to outperform SGD (designed for generic stochastic problems) and N-SAGA (designed for related settings). This setting subsumes important applications where data augmentation\perturbation have been proven successful (particularly, image and text classification), rendering it meaningful and interesting. On the flip side, the convergence analysis is somewhat loosely stated (e.g., the use of epsilon bar) and there are highly non-trivial user-specified parameters which have to be determined and carefully tuned. Moreover, although the expected performance on multi-layer network are considered in the body of the paper, they are not addressed appropriately in Section 4 where supporting experiments are presented (only 2-layers networks) - I find this very disappointing as this one of the main promises of the paper. Lastly, the paper is relatively easy-to-follow. Comments ======== L8 - As convergence rates are usually stated using O-notation, it is not utterly clear how to inrepret 'smaller constant factor'. L17 - What does 'simple' stand for? proximal-friendly? L21 - Can you provide pointers\cross-reference to support 'fundamental'? Also, consider rephrasing this sentence. L47 - Consider repharsing sentence. L65 - Table 2: Data radius is not seemed to be defined anywhere in the text. L66 - How does concentrating on a quantity that depends only on the minimizer affect the asymptotic properties of the stated convergence analysis? L69 - 'note that\nabla' is missing a space. L77 - In what sense is this rate 'unimprovable'? L78 - This pararaph seems somewhat out of place. Maybe put under 'realted work'? L90 - This paragrpah too seems somewhat out of place (seems to be realted to section 4). L116+L121 - the wording 'when f_i is an expectation' is somewhat confusing. L183+L185 - The use of epsilon bar is not conventional and makes it hard to parse this statement. Please consider restating the convergence analysis in terms of the pre-defined accuracy level (alterntaively, consider providing more explenatation of the upper bound parameters). L192 - Can you elaborate more on concrete choices for gamma? L198 - 'as large as allowed' in terms of EQ 9? L200 - Figure 1: can you elaborate more on 'satisfies the thoery'? L208 - some expremients seem to be 400-epochs long. L226 - Here and after, numbers are missing thousands separator. L261 - Comparing could be more fair if a non-uniform version of SGD would be used. L271 - Line seems out of place.